nanotechnology

graphene quantum dots, silver nanoparticles, humidity sensor, Schottky junction

**Author for correspondence:**
Chatchawal Wongchoosuk
e-mail: chatchawal.w@ku.ac.th

This article has been edited by the Royal Society of Chemistry, including the commissioning, peer review process and editorial aspects up to the point of acceptance.

# High-performance resistive humidity sensor based on Ag nanoparticles decorated with graphene quantum dots

Gun Chaloeipote[1], Jaruwan Samarnwong[1],
Pranlekha Traiwatcharanon[1], Teerakiat Kerdcharoen[2] and
Chatchawal Wongchoosuk[1]

[1]Department of Physics, Faculty of Science, Kasetsart University, Chatuchak, Bangkok 10900, Thailand
[2]Department of Physics, Faculty of Science, Mahidol University and Research Network of NANOTEC at Mahidol University, National Nanotechnology Center, Bangkok 10400, Thailand

CW, 0000-0002-5613-6615

In this work, we present a low-cost, fast and simple fabrication of resistive-type humidity sensors based on the graphene quantum dots (GQDs) and silver nanoparticles (AgNPs) nanocomposites. The GQDs and AgNPs were synthesized by hydrothermal method and green reducing agent route, respectively. UV–Vis spectrophotometer, X-ray photoelectron spectroscopy and field-emission transmission electron microscopy were used to characterize quality, chemical bonding states and morphology of the nanocomposite materials and confirm the successful formation of core/shell-like AgNPs/GQDs structure. According to sensing humidity results, the ratio of GQDs/AgNPs 1 : 1 nanocomposite exhibits the best humidity response of 98.14% with exponential relation in the humidity range of 25–95% relative humidity at room temperature as well as faster response/recovery times than commercial one at the same condition. The sensing mechanism of the high-performance GQDs/AgNPs humidity sensor is proposed via Schottky junction formation and intrinsic synergistic effects of GQDs and AgNPs.

## 1. Introduction

Graphene quantum dots (GQDs) have attracted worldwide research attention in the past few years. The GQDs are zero-dimensional inorganic semiconductor nanomaterials having the diameter size below 20 nm [1]. According to quantum confinement effects, the GQDs own a non-zero bandgap, luminesce on excitation, unique

optical properties, large surface/volume ratio, tunable electronic properties by sizes, high active sites, low toxicity and photoluminescence [2] leading to diverse applications in bioimaging [3], energy storage [4], fuel cells [5], optical devices [6], drug delivery [7] and sensors [8]. The GQDs can be generally synthesized by two main approaches, namely bottom-up and top-down methods. The bottom-up method is based on the molecular precursors by chemical reactions [9]. The top-down process is another method that is appropriated for macroscopic scale because of the simple preparation process such as laser ablation [10], electrochemical oxidation [11], chemical ablation or oxidation [12] and ultrasonication [13].

Silver nanoparticles (AgNPs) are inorganic materials presenting dominant chemical, optical and physical properties which are controlled by their nature shape and size. The particles exhibit high conducting electron density and the appearance of surface plasmon resonances [14]. As a result, they are applied in different fields such as antibacterial, medicine, photonics, sensor and catalysis [15–17]. In general, hazard chemical or non-biodegradable agents were used for reducing Ag ion into AgNPs. To protect environment and reduce the production cost, green synthesis becomes an alternative method to synthesize AgNPs because it provides eco-friendly, cheap, biocompatible and ease to perform based on reducing agent by microorganisms, enzymes, bacteria, fungus and plants [18–23].

Humidity sensors play a crucial role in various fields such as agriculture [24], medical [25], climatology [26], food processing [27] and household appliances [28]. One of the many types of humidity sensor is based on capacitive working mechanism. However, the capacitive humidity sensor needs complicated electronic parts and data processing. The resistive type may be a better choice for next generation of humidity sensor due to simple fabrication, cost effectiveness, tiny size and flexible properties [29]. The performance of resistive-type humidity sensor relies on the highly resistive sensing material deposited on interdigitated electrodes (IDEs). Nowadays, many researchers are interested in developing sensing performances of humidity sensors by selection of various sensing materials such as polymers, ceramic and nanostructured metal oxides [30–34] to gain good stability, longer lifetime, wide range of humidities, high responsibility, fast response and recovery times. For examples, Arena *et al.* [35] studied mixture of diallyldimethylammonium chloride and nanosized $Fe_2O_3$ powder by deposition on multi-walled carbon nanotubes electrodes exposure to relative humidity level in the range of 35–60%. Yao *et al.* [36] fabricated graphene oxide (GO)-based humidity sensor deposited on quartz crystal microbalances and revealed a wide range of humidities response (6.4–93.5% relative humidity (RH)). Su *et al.* [37] prepared humidity sensor based on combination of gold nanoparticles and GO by sol–gel technique which showed good response of RH in the range 20–90%. Malik *et al.* [38] synthesized In-$SnO_2$-loaded mesoporous g-CN nanocomposites that showed excellent response and high stability to humidity in 11–98% RH. Singh *et al.* [39] introduced new Truxene-based covalent organic framework for good humidity sensing application. Tomer *et al.* [40] fabricated a high-performance humidity sensor based on mesoporous g-CN/AgNPs which demonstrated very fast response time (3 s) and recovery time (1.4 s) to a wide %RH range (11–98% RH). However, based on our best knowledge, no report about combination between GQDs and AgNPs nanocomposite for humidity sensing properties is available.

In this approach, based on the benefits of GQDs and AgNPs, we have studied a corporation between the inorganic semiconductors (GQDs) into metal nanoparticles (AgNPs) for humidity sensing application. The GQDs were synthesized by simple hydrothermal method and AgNPs were prepared by green reducing agent synthesis. The resistive humidity sensors based on GQDs/AgNPs nanocomposites were fabricated by drop-casting onto IDEs. The sensing characteristic responses were systematically investigated in the range of 25–95% RH at room temperature. The effects of varied compositions and sensing mechanism will be discussed in detail.

# 2. Experimental section

## 2.1. Synthesis of graphene quantum dots

A schematic of the GQDs synthesis steps based on a modified hydrothermal method [41] is shown in figure 1*a*. Forty milligrams of black solid GO powder purchased from ACS material was dispersed in 40 ml deionized (DI) water under mild stirring for 15 min. Afterwards, the dispersed GO was exfoliated by ultrasonication for 1.5 h at room temperature. Two hundred microlitres of ammonia were then added into GO solution under mechanical stirring for 15 min at room temperature in order to slice and stabilize GO into GQDs. The solution was transferred to Teflon-lined stainless-steel autoclave and heated at 140°C for 13 h. After the autoclave was cooled to room temperature, the obtained precipitate was centrifuged for 45 min and washed three times with ethanol and DI water.

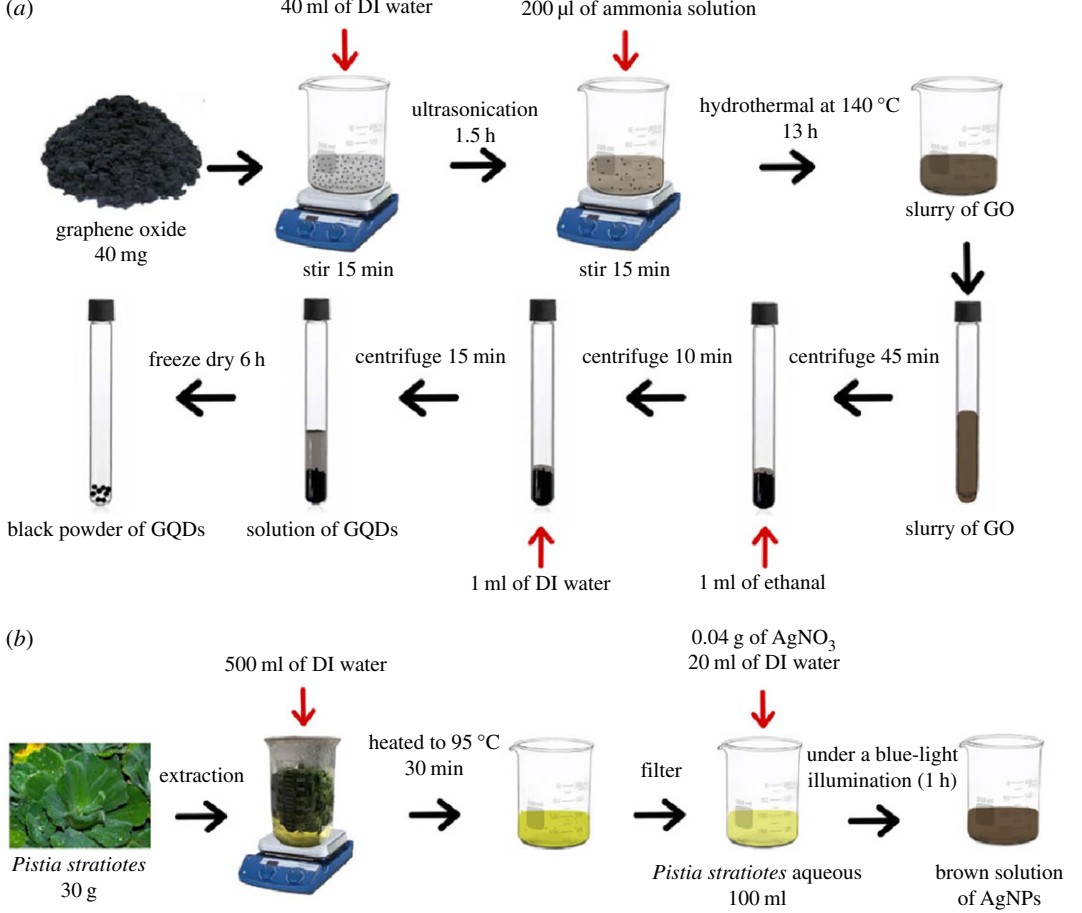

**Figure 1.** Schematic diagram for the synthesis of (*a*) GQDs and (*b*) AgNPs.

The remaining solid material was then freeze dried for 6 h. The black powder of GQDs was received and kept at room temperature for further use.

## 2.2. Synthesis of silver nanoparticles

The AgNPs were synthesized by a simple one-pot green route method as shown in figure 1*b*. The fresh leaves of *Pistia stratiotes* (30 g) were cut into small pieces and cleaned by DI water. Then, they were heated in 500 ml of DI water at 95°C for 30 min and cooled down to room temperature. The reducing solution was filtered by Whatman® Grade 1 filter paper. Next, 0.04 g of silver nitrate (AgNO₃) in 20 ml DI water was filled into the reducing *P. stratiotes* solution (100 ml). To reduce Ag ion into AgNPs, the mixture solutions were put under a blue-light illumination (8 W lamp) for 1 h in a dark chamber at room temperature. Successful formation of AgNPs can be observed by colour change from yellow to brown with the naked eye.

## 2.3. Fabrication of GQDs/AgNPs humidity sensors

The obtained samples were dispersed in 1 ml of DI water. The GQDs/AgNPs humidity sensors were fabricated by simple drop-casting method as shown in figure 2*a*. The GQDs and AgNPs were mixed by various volume ratio of 4 : 1, 3 : 2, 1 : 1, 2 : 3 and 1 : 4, respectively, under ultrasonication for 30 min. Silver conductive paint IDEs with the size of $1 \times 1.5$ cm and spacing 1 mm were screened on transparent plastic substrate (polyethylene terephthalate). Then, the mixed aqueous solutions were deposited on electrodes by drop-casting (30 µl) and dried in air at room temperature.

## 2.4. Characterization

The absorption spectra of the synthesized samples were recorded by Shimadzu 1800 UV–Vis spectrophotometer in the absorption wavelength of 200–500 nm. Elemental composition and chemical

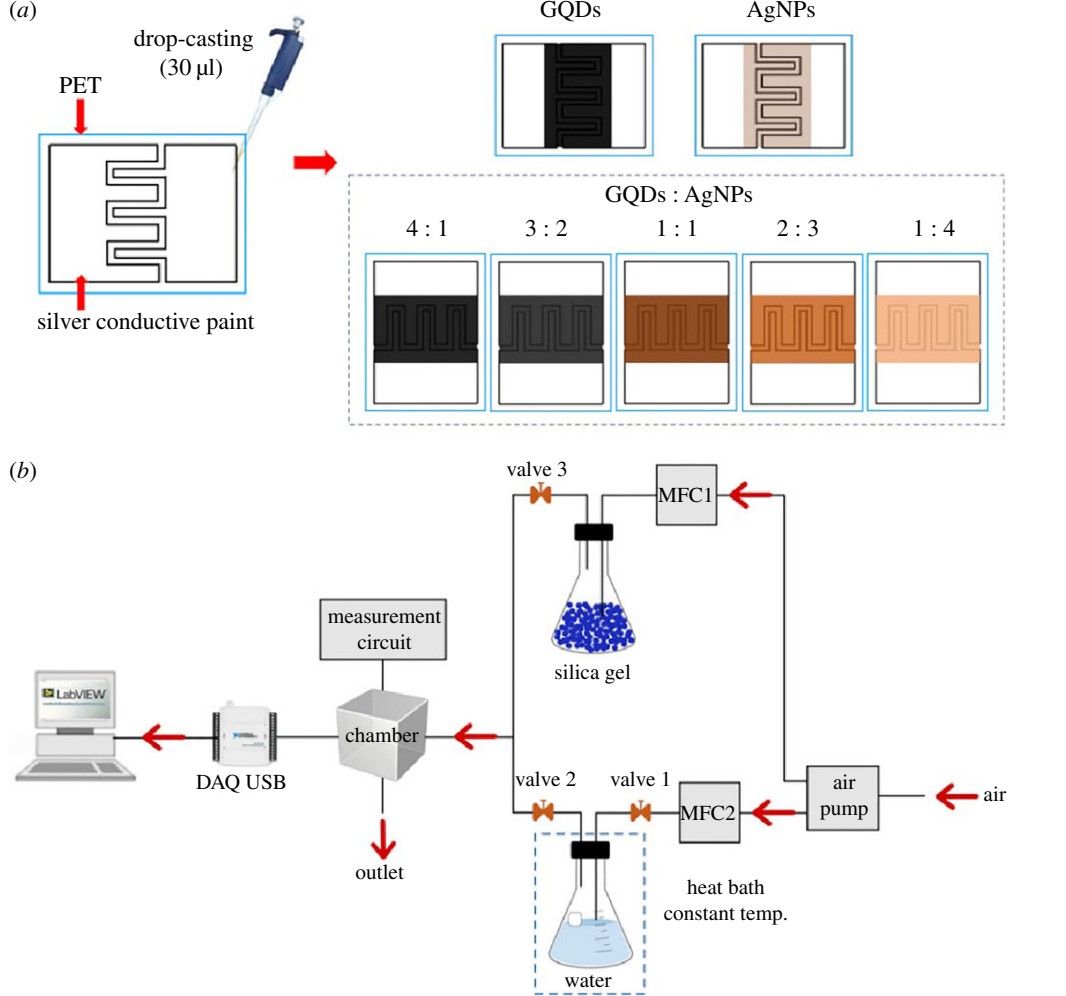

**Figure 2.** (*a*) Humidity sensor fabrications and (*b*) schematic diagram for humidity sensor measurement.

bonding states of samples were examined via Axis Ultra DLD X-ray photoelectron spectroscopy (XPS). Field-emission transmission electron microscopy (FE-TEM) model JEM-3100 (HR) was used to investigate the morphology. The size distribution of materials was calculated using Image J software.

## 2.5. Measurement of humidity sensing

The measurements of humidity sensing were performed using a standard flow-through system with our designed 36 ml Teflon chamber as shown in figure 2*b*. The water vapour generated from a bubbling system was used as a source of humidity and placed in a temperature-controlled heat bath. The RH range from 25 to 95% was varied by adjusting two mass flow controllers (MFC1 and MFC2). The mass flow ratios to obtain a desired %RH are listed in electronic supplementary material, table S1. The total flow rate (MFC1 + MFC2) was fixed at 200 standard cubic centimetres per minute (SCCM). The measurement was carried out at room temperature (25.0 ± 1°C). The real-time RH and temperature changes were measured by commercial humidity and temperature (SHT15) sensors embedded in the sensor chamber. The measurement started from a flow of dry air into the test chamber for 2 min (baseline of sensors). Then, humidified air generated from the water flask was carried into the chamber for 3 min for humidity testing. The electrical characteristics of sensors were measured by a constant applied voltage (10 V) based on a simple voltage divider circuit. The data were collected every second using our developed LabVIEW software via a USB DAQ device (NI USB-6008) for subsequent analyses. The humidity sensing response (%) of the sensors can be calculated as follows:

$$S_{\text{response}}(\%) = \left| \frac{R_{\text{humidity}} - R_{\text{dry}}}{R_{\text{dry}}} \right| \times 100, \tag{2.1}$$

where $R_{\text{humidity}}$ and $R_{\text{dry}}$ are the resistance of sensor in humidified air and dry air (25% RH), respectively.

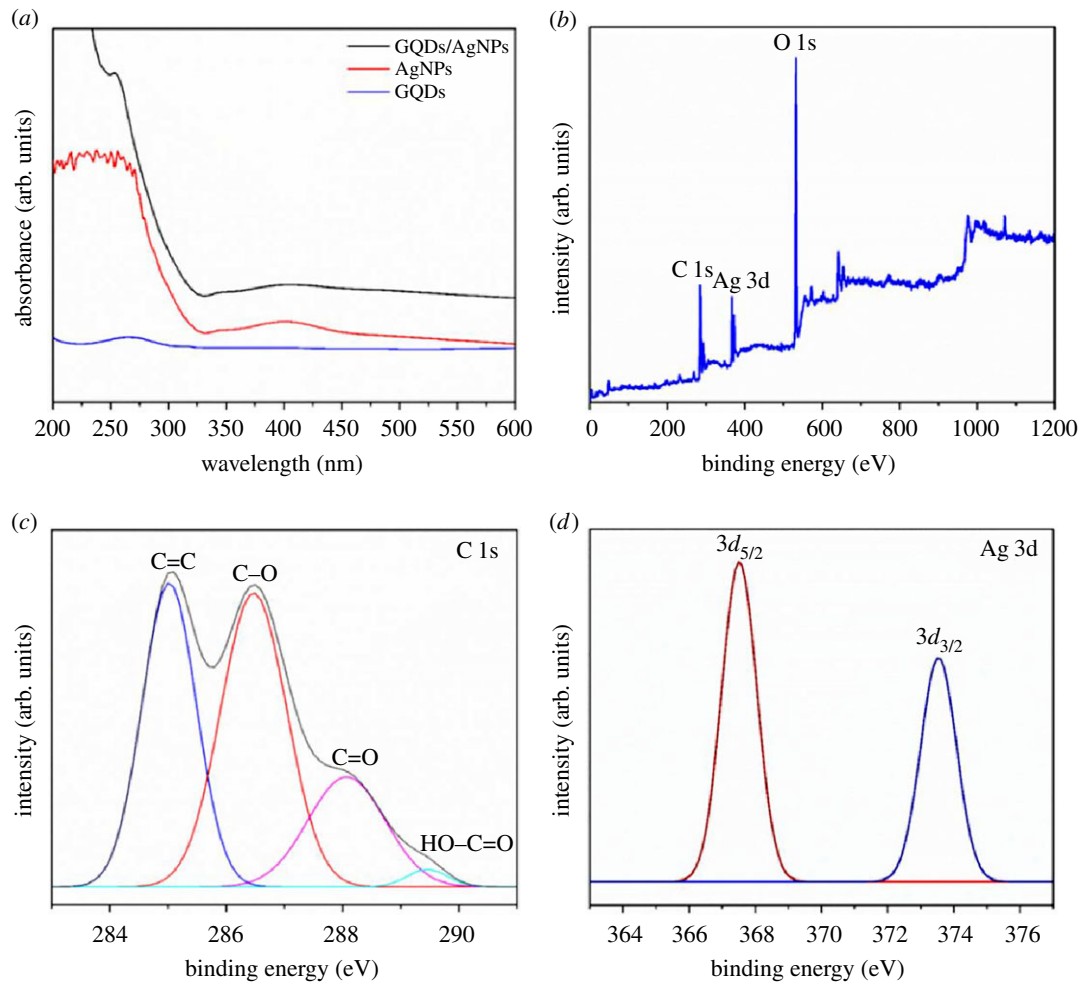

**Figure 3.** (*a*) UV–Vis spectra of GQDS, AgNPs and GQDs/AgNPs nanocomposite. (*b*) XPS survey scans, (*c*) C 1s and (*d*) Ag 3d core-level spectrum of GQDs/AgNPs nanocomposite.

# 3. Results and discussion

## 3.1. Characterization of synthesized nanomaterials

UV–Vis spectra of GQDs/AgNPs nanocomposite show the typical absorption peaks of both GQDs and AgNPs at approximately 263 nm and approximately 400 nm (figure 3*a*) which attributed to the $\pi \rightarrow \pi^*$ transition of $sp^2$ carbon bonds [42] and complete formation of the silver ions into AgNPs [20], respectively. The formation mechanism of GQDs from GO is based on the etching and reduction of $OH^-$ generated from $NH_3$ dissociation in water [43]. The $OH^-$ reacts with defect sites of GO and breaks down GO into small fragments. In the case of AgNPs formation, the polyphenols from *P. stratiotes* extract help to reduce $Ag^+$ to $Ag^0$ via phenolic form leading to aggregate into Ag clusters and AgNPs [20]. Chemical composition of GQDs/AgNPs can be identified via XPS spectra, as shown in figure 3*b–d*. The wide region XPS survey scan (figure 3*b*) clearly shows spectra of C 1s, Ag 3d and O 1s peaks. The high-resolution C 1s spectrum (figure 3*c*) consists of four peaks with the binding energies at 285.0, 286.5, 288.1 and 289.5 eV related to C=C, C–O, C=O and HO–C=O, respectively. The high-resolution Ag (3d) spectrum (figure 3*d*) displays two intense peaks with the binding energies at 367.5 and 373.5 eV assigned to spin–orbit coupling of $3d_{5/2}$ and $3d_{3/2}$, respectively. The results indicate that AgNPs present in the metallic form and the GQDs contain the oxygen and hydroxyl functional groups.

To clarify the morphologies of synthesized nanomaterials, the TEM images of the GQDs, AgNPs and GQDs/AgNPs nanocomposite are displayed in figure 4. As illustrated in figure 4*a*, the GQDs show the formation of mono-distribution with spherical shape corresponding to $OH^-$ generated from ammonia as excellent reducing and stabilizing agent. Their diameters are in the range of 1.5–3.5 nm (2.8 nm average

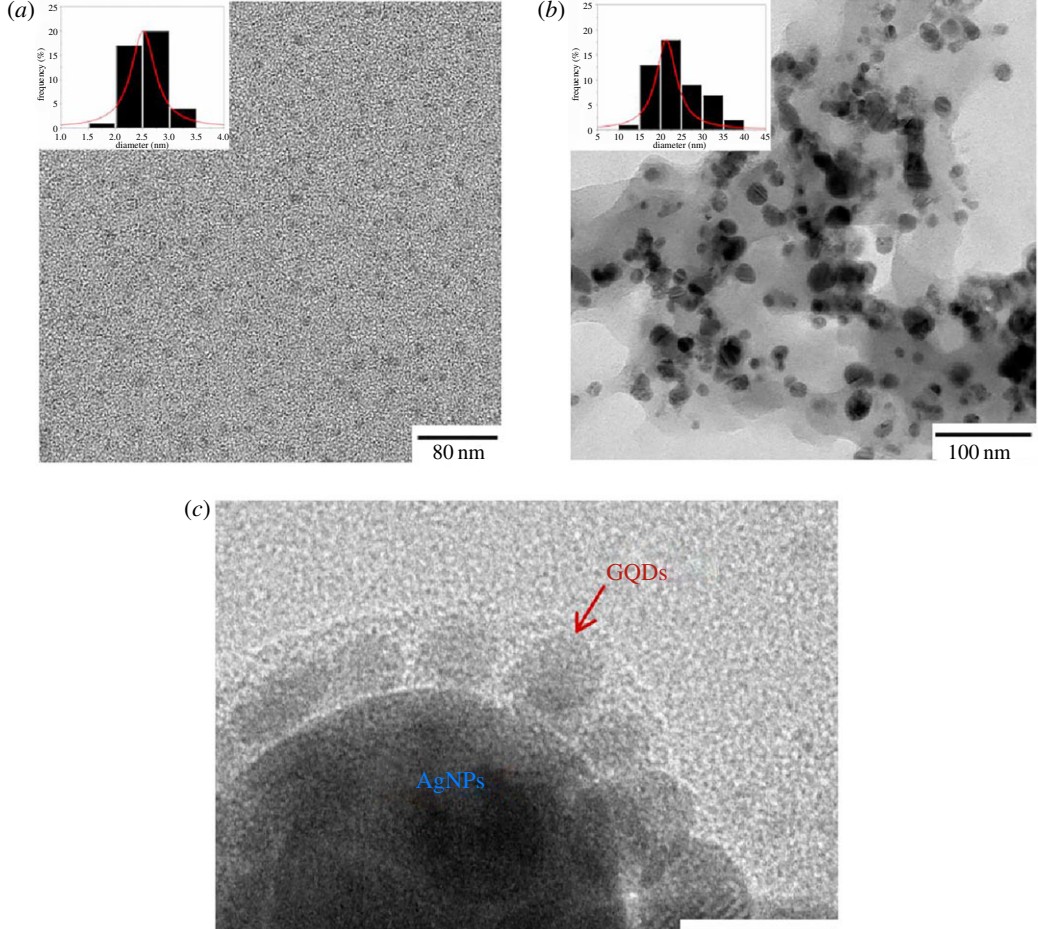

**Figure 4.** TEM images of (*a*) GQDs, (*b*) AgNPs and (*c*) GQDs/AgNPs nanocomposite.

diameter), as shown in the histogram in the inset of figure 4*a*. For pristine AgNPs, they exhibit agglomeration of spherical particles with uniform size. The AgNPs diameters are in the range of 10–40 nm (22 nm average diameter) as shown in the histogram in the inset of figure 4*b*. To combine the GQDs and AgNPs by simple normal solution-based mixture under ultrasonication for 30 min, the AgNPs/GQDs nanocomposite exhibits a core/shell-like structure as shown in figure 4*c*. The GQDs are well decorated on the AgNPs surface that can present Schottky junctions between sensing materials and may provide a high specific surface area and active sites for the water molecules adsorption.

## 3.2. Humidity sensing performance

The humidity sensing performance of the GQDs/AgNPs (1 : 1) nanocomposite sensor towards different RH from 25% RH (baseline) to 95% RH upon five cycles is displayed in figure 5*a*. The electrical resistance of the sensor rapidly decreases upon the water adsorption and completely recovers to the initial stage after water desorption at the dry air (25% RH). The calculated humidity response of GQDs/AgNPs (1 : 1) sensor is approximately 98.14% with standard deviation (±0.94%) upon five cycles suggesting high response and excellent repeatability of the %RH detection. To investigate the effect of GQDs/AgNPs ratio, the GQDs/AgNPs humidity sensors with various volume ratios of 4 : 1, 3 : 2, 1 : 1, 2 : 3 and 1 : 4 as well as the pristine GQDs and pristine AgNPs were tested at the same %RH condition, as depicted in figure 5*b*. All sensors exhibit the similar humidity sensing behaviour in which the electrical resistances decrease with increasing %RH from 25 to 95%. The pristine GQDs and AgNPs sensors show the humidity response of approximately 70.46% and 84.35%, respectively. The humidity responses of GQDs/AgNPs 4 : 1, 3 : 2, 2 : 3 and 1 : 4 nanocomposites are approximately 72.72%, 85.62%, 92.5% and 87.02%, respectively. The GQDs/AgNPs (1 : 1) sensor clearly shows the highest humidity sensing performance. This result may be attributed from the different electron mean free

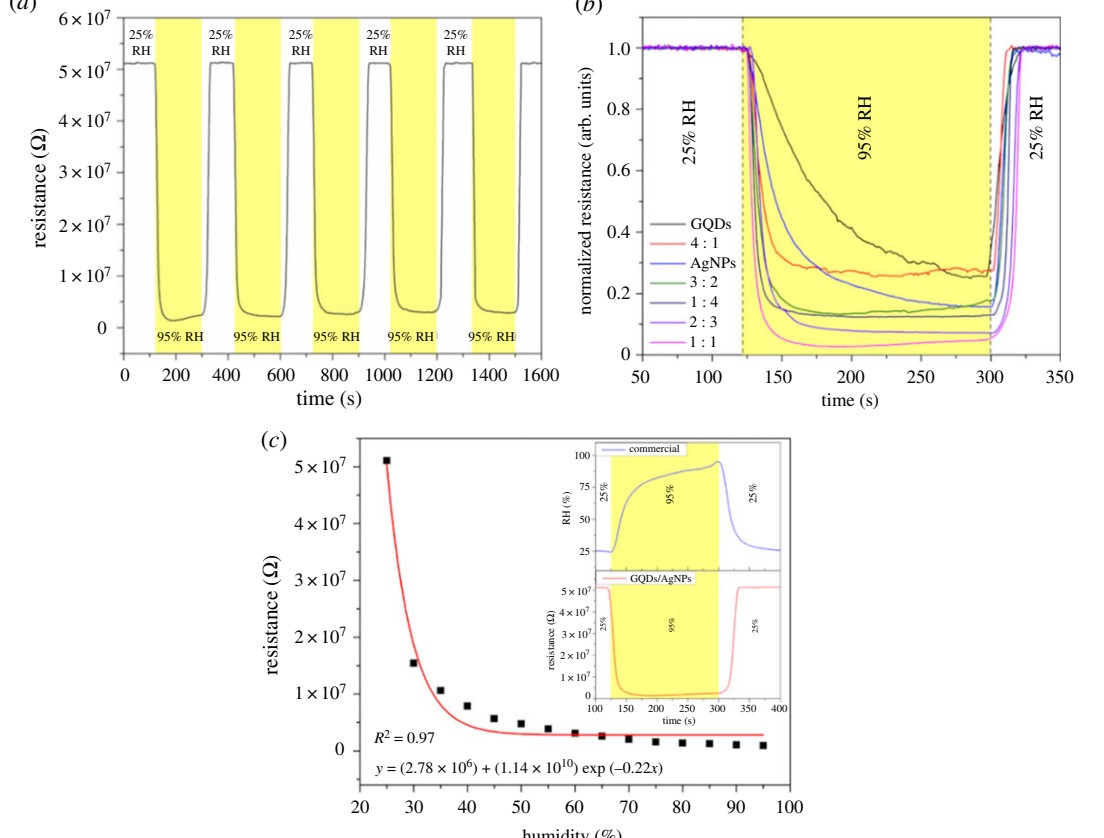

**Figure 5.** Humidity dynamic response of (a) the GQD/AgNPs (1:1) sensor and (b) various ratio GQD/AgNPs sensors as well as pristine GQD and pristine AgNPs. (c) Resistance of GQD/AgNPs as a function of RH, and inset shows a humidity dynamic response of GQD/AgNPs (1:1) sensor compared with a commercial humidity SHT-15 sensor.

path [44] and catalytic action [45]. Electrons may be localized on the AgNPs and promoted the high response after adding the electronic balance of GQDs. As a result, the total value of resistance decreases and the response increases [46]. However, more composition of AgNPs tends to show lower response for 2:3 and 1:4. It should be indicated that the addition of Ag ratio has a limit because AgNPs conductivity in the composited sensor are too high, the initial resistance of the composites shows too small value to be sensing material described by percolation thresholds [47].

An exponential fit was plotted for the correlation between the resistance and the humidity concentrations as demonstrated in figure 5c. The average resistance values of GQDs/AgNPs sensor in dry air (25% RH), 35%, 45%, 55%, 65%, 75%, 85% and 95% RH are approximately 51.1, 10.6, 5.67, 3.87, 2.57, 1.60, 1.26 and 0.95 MΩ, respectively. Other values can be found from the datasets, and the enlarged graph for clear observation at high humidity is displayed in electronic supplementary material, figure S1. The equation is expressed as $y = (2.78 \times 10^6) + (1.14 \times 10^{10}) \exp(-0.22x)$, where $y$ is the resistance and $x$ is % RH. The correlation coefficient ($R^2$) was calculated to be 0.97. The resistance of humidity sensor follows a good exponential response relationship in the range of 25–95% RH. It may be caused by contributions of the Schottky barrier modulation rather than ohmic contact upon water molecule adsorption. After 95% RH, the resistance trends to observe saturation behaviour. It may be due to the saturation appearance on the surface of the sensing material with maximum amount of water molecules. This phenomena can be related to the capillary condensation in nanoparticles between 2 and 100 nm [48]. To compare the performance of the GQD/AgNPs sensor for humidity detection, a list of sensing nanomaterials based on humidity sensors is summarized in table 1. The GQD/AgNPs sensor in the present work is comparable and superior to previous publications. Moreover, we have compared the humidity sensing performance of the GQD/AgNPs sensor with the popular commercial sensor (SHT-15) as shown in an inset of figure 5c. The commercial sensor shows the response and recovery times of 25 s and 30 s, respectively. In the case of GQDs/AgNPs nanocomposite, both response and recovery times of GQD/AgNPs sensor are 15 s. The

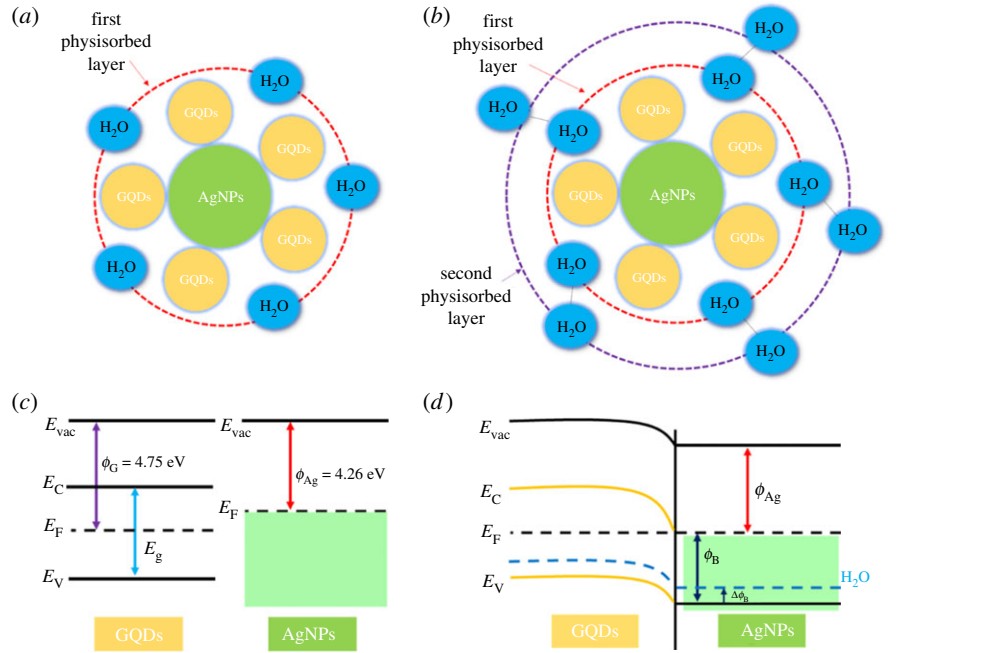

**Figure 6.** Sensing mechanism illustrations of (*a*) the first and (*b*) the second physisorbed layer of the water molecules. (*c*) Energy band diagrams of GQDs and AgNPs before contact and (*d*) formation of Schottky barrier with/without $H_2O$ adsorption.

**Table 1.** List of sensing nanomaterials for humidity detection.

| sensor material | target | structures | response time (s) | % RH | sensing temp. (°C) | response (%) | refs. |
|---|---|---|---|---|---|---|---|
| GQDs/AgNPs | $H_2O$ | nanocomposite | 15 | 95 | RT | 98.14 | this work |
| GO | $H_2O$ | thin film | 45 | 93.5 | RT | 22.1 | [36] |
| AuNPs/GO/MPTMOS | $H_2O$ | sol–gel film | 119 | 90 | RT | — | [37] |
| SnO₂/PANI | $H_2O$ | nanocomposite | 26 | 95 | RT | 90 | [49] |
| Sb–SnO₂ | $H_2O$ | nanowire | — | 40 | RT | 1.4 | [50] |
| rGO/MoS₂ | $H_2O$ | hybrid composites | 17 | 50 | RT | 49 | [29] |

GQD/AgNPs sensor clearly shows a faster response and recovery time at the same conditions. It should be noted that the response and recovery times were calculated as the time to reach 90% of the final equilibrium value [51].

## 3.3. Humidity sensing mechanism

The sensing mechanism of the GQDs/AgNPs nanocomposite sensor for humidity detection at room temperature relies on the physical adsorption. At low RH level (25% RH), water molecules are absorbed onto the available active sites of GQDs/AgNPs as the primary physisorption layer of water corresponding to double hydrogen bonding as shown in figure 6*a*. The water molecules cannot freely move and no proton conduction occurs due to the constraint from double hydrogen bonding. Consequently, GQDs/AgNPs mainly present intrinsic conductance behaviour (high resistance). As the RH content increases, the secondary physisorption layer of water molecules appears as displayed in figure 6*b*. The liquid-like behaviour is gradually observed and the protons can move freely. Moreover, the water in the outer physisorbed layer can be ionized to hydronium ions as a charge carrier which can hop between adjacent water molecules and generate Grotthuss chain reaction [52]. All these factors lead to a high electrical conductivity of GQDs/AgNPs sensing material at high RH.

High humidity sensing response of GQDs/AgNPs over pristine GQDs and pristine AgNPs may be attributed to two main aspects: (i) formation of Schottky junctions at the metal–semiconductor

interfaces that play a crucial role for charge transfer enhancements [53,54]. Energy band diagrams of GQDs and AgNPs before contact are displayed in figure 6c. The work functions of GQD ($\phi_G$) and Ag ($\phi_{Ag}$) are approximately 4.75 and approximately 4.26 eV, respectively [55,56]. When GQD are well embedded on AgNPs, the electrons can be transferred between two sensing nanomaterials to equilibrate the Fermi energy level. The band bending occurs and forms Schottky barrier [53,54] (figure 6d). At low RH level (25% RH), Schottky barrier ($\phi_B$) exhibits a high value based on the Schottky junction formation with intrinsic conductions of metal–semiconductor nanocomposites. At high RH levels, the absorbed water molecules can attract electrons on the surface resulting to decrease Schottky barrier ($\Delta\phi_B$) as shown in figure 6d (blue dot line). Therefore, it causes a big change in the sensor resistance/electrical conductivity leading to high sensitivity compared with pristine GQDs or pure AgNPs. (ii) High humidity sensing performance of GQDs/AgNPs sensor comes from intrinsic synergistic effects of GQDs and AgNPs for water adsorption. It is well known that Ag is a good oxygen adsorption catalyst which is able to chemisorb and dissociate $O_2$ under atmospheric conditions while GQDs present oxygen and hydroxyl functional groups to increase hydrogen bonding and interactions with water molecules resulting to high response of sensor.

# 4. Conclusion

We have demonstrated the room-temperature resistive humidity sensors based on GQDs/AgNPs nanocomposites deposited on interdigitated Ag electrodes by drop-casting method. The UV–Vis, XPS and FE-TEM characterization results confirmed the successful decoration of GQDs around the surface of AgNPs to form metal–semiconductor interfaces. According to the Schottky junction formation and intrinsic synergistic effects of the GQDs/AgNPs nanocomposites, the humidity sensitivity of GQDs/AgNPs (1:1) presented the highest relative response of 98.14% with an exponential relation in the range of 25–95% RH. The response and recovery times were estimated 15 s as faster than the commercial one (SHT15) at the same conditions. The sensing mechanism was described based on physical adsorption and metal–semiconductor-material-based humidity sensor via the Schottky junctions. Based on the results, we thoroughly believe that the GQDs/AgNPs nanocomposites will become one of the best humidity sensing nanomaterials for next generation of resistive-type humidity sensors.

Data accessibility. Datasets and supplementary information are available from the Dryad Digital Repository: https://doi.org/10.5061/dryad.1g1jwstvq [57].

Authors' contributions. G.C. performed the experiments, analysed results and drafted the manuscript; J.S. prepared the graphics; P.T. carried out the synthesis experiment; T.K. participated in data analysis; C.W. conceived of the study, designed the study, coordinated the study, analysed the results, supervised the project and helped draft the manuscript. All authors gave final approval for publication and agree to be held accountable for the work performed therein.

Competing interests. We declare we have no competing interests.

Funding. This research was supported by Kasetsart University Research and Development Institute (KURDI) under the grant no. FF(KU) 25.64.

Acknowledgements. C.W. acknowledges Kasetsart University Research and Development Institute for financial support of this work. G.C. acknowledges the scholarship from Science Achievement Scholarship of Thailand (SAST).

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
