## [Peer Review File · Royal Society Open Science]

Review History

RSOS-210407.R0 (Original submission)

Review form: Reviewer 1

Is the manuscript scientifically sound in its present form?

Yes

Are the interpretations and conclusions justified by the results?

Yes

Is the language acceptable?

Yes

Do you have any ethical concerns with this paper?

Yes

Have you any concerns about statistical analyses in this paper?

No

Recommendation?

Accept with minor revision (please list in comments)

Comments to the Author(s)

In the manuscript RSOS-210407, the authors have hydrothermally prepared graphene quantum dots (GQDs) and silver nanoparticles (AgNPs) core-shell nanocomposite based resistive %RH sensors. A variety of characterization techniques have been used to analyze the sensing material, while several sensing attributes including sensitivity, reversibility, response/recovery time etc, were evaluated. The manuscript is interesting, good for publication and within the scope of the Royal Society Open Science Journal. There are following minor concerns:

1. "The GQDs are zero-dimension inorganic semiconductor...". It should be 'dimensional'.
2. "One of the most types of humidity sensor is based on capacitive.". The sentence seems incomplete.
3. "...longer lifetime, high responsibility, rapid response and recovery time and wider humidity detection range.". Authors have discussed about 'wider %RH range' here, however in the examples they have provided detection ranges to be 35-60% and 20-90%. Please cite relevant examples with wide %RH range. E.g. 10.1039/C7TA02860A; 10.1039/C6NR08039A; 10.1039/C7TA05043G
4. "The GQDs were synthesized by simple top-down via hydrothermal method and..." It is better to remove the word 'top-down' from here.
5. What is the standard deviation in Figure 5a
6. Important %RH sensor based reference can be cited: doi.org/10.1063/1.5123479

Review form: Reviewer 2

Is the manuscript scientifically sound in its present form?

Yes

Are the interpretations and conclusions justified by the results?

Yes

Is the language acceptable?

Yes

Do you have any ethical concerns with this paper?

No

Have you any concerns about statistical analyses in this paper?

No

Recommendation?

Major revision is needed (please make suggestions in comments)

Comments to the Author(s)

Authors have studied humidity sensing properties of Ag/GQDs composite in details. However, there are few grammatically mistakes and poorly constructed sentences. I would suggest authors to check manuscript carefully.

1. How did you vary humidity? explain in detail in the experimental section.

2. What is flow rate in SCCM of air entering in the sensing chamber and humid air. Does it vary for different humidity testing? Flow rate affects sensor's resistance and sensing performance. Ideally flow rate should constant for all measurement.
3. Please write resistance values of sensor in dry air and different %RH in manuscript.
4. Ideally sensors response should be linear but in your case change in response is not significant after >50%RH. what could be reason.
5. Letter C in the temperature unit should be in capital. please correct figure 1 and also correct in manuscript.
6. Please provide references to mechanism explained in figure 6. Also work function of graphene shown in figure 6 is incorrect. Draw initial fermi levels of both GQDs and Ag and how band bending occurs and forms Schottky barrier. It would be great if you add it in figure 6.

Decision letter (RSOS-210407.R0)

Dear Professor Wongchoosuk:

Title: High-Performance Resistive Humidity Sensor Based on Ag Nanoparticles Decorated with Graphene Quantum Dots
Manuscript ID: RSOS-210407

The editor assigned to your manuscript has now received comments from reviewers. We would like you to revise your paper in accordance with the referee and Subject Editor suggestions which can be found below (not including confidential reports to the Editor). Please note this decision does not guarantee eventual acceptance.

Please submit your revised paper before 04-Jun-2021. Please note that the revision deadline will expire at 00.00am on this date. If we do not hear from you within this time then it will be assumed that the paper has been withdrawn. In exceptional circumstances, extensions may be possible if agreed with the Editorial Office in advance. We do not allow multiple rounds of revision so we urge you to make every effort to fully address all of the comments at this stage. If deemed necessary by the Editors, your manuscript will be sent back to one or more of the original reviewers for assessment. If the original reviewers are not available we may invite new reviewers.

On behalf of the Subject Editor Professor Anthony Stace and the Associate Editor Dr Dattatray Late.

RSC Associate Editor:
Comments to the Author:
Major Revision

RSC Subject Editor:
Comments to the Author:
(There are no comments.)

Reviewers' Comments to Author:
Reviewer: 1

Comments to the Author(s)

In the manuscript RSOS-210407, the authors have hydrothermally prepared graphene quantum dots (GQDs) and silver nanoparticles (AgNPs) core-shell nanocomposite based resistive %RH sensors. A variety of characterization techniques have been used to analyze the sensing material, while several sensing attributes including sensitivity, reversibility, response/recovery time etc, were evaluated. The manuscript is interesting, good for publication and within the scope of the Royal Society Open Science Journal. There are following minor concerns:

1. "The GQDs are zero-dimension inorganic semiconductor...". It should be 'dimensional'.
2. "One of the most types of humidity sensor is based on capacitive.". The sentence seems incomplete.
3. "...longer lifetime, high responsibility, rapid response and recovery time and wider humidity detection range.". Authors have discussed about 'wider %RH range' here, however in the examples they have provided detection ranges to be 35-60% and 20-90%. Please cite relevant examples with wide %RH range. E.g. 10.1039/C7TA02860A; 10.1039/C6NR08039A; 10.1039/C7TA05043G
4. "The GQDs were synthesized by simple top-down via hydrothermal method and..." It is better to remove the word 'top-down' from here.
5. What is the standard deviation in Figure 5a
6. Important %RH sensor based reference can be cited: doi.org/10.1063/1.5123479

Reviewer: 2

Comments to the Author(s)

Authors have studied humidity sensing properties of Ag/GQDs composite in details.

However, there are few grammatically mistakes and poorly constructed sentences. I would suggest authors to check manuscript carefully.

1. How did you vary humidity? explain in detail in the experimental section.
2. What is flow rate in SCCM of air entering in the sensing chamber and humid air. Does it vary for different humidity testing? Flow rate affects sensor's resistance and sensing performance. Ideally flow rate should constant for all measurement.
3. Please write resistance values of sensor in dry air and different %RH in manuscript.
4. Ideally sensors response should be linear but in your case change in response is not significant after >50%RH. what could be reason.
5. Letter C in the temperature unit should be in capital. please correct figure 1 and also correct in manuscript.
6. Please provide references to mechanism explained in figure 6. Also work function of graphene shown in figure 6 is incorrect. Draw initial fermi levels of both GQDs and Ag and how band bending occurs and forms Schottky barrier. It would be great if you add it in figure 6.

Author's Response to Decision Letter for (RSOS-210407.R0)

See Appendix A.

Decision letter (RSOS-210407.R1)

Dear Professor Wongchoosuk:

Title: High-Performance Resistive Humidity Sensor Based on Ag Nanoparticles Decorated with Graphene Quantum Dots

Manuscript ID: RSOS-210407.R1

It is a pleasure to accept your manuscript in its current form for publication in Royal Society Open Science. The chemistry content of Royal Society Open Science is published in collaboration with the Royal Society of Chemistry.

On behalf of the Subject Editor Professor Anthony Stace and the Associate Editor Dr Dattatray Late.

RSC Associate Editor
Comments to the Author:
Authors have addressed all the comments. Now manuscript is suitable for publication.

Reviewer(s)' Comments to Author:

Appendix A

RESPONSE TO REVIEWERS

Reviewer # 1:

Thank you for your review and valuable recommendation. The paper has been revised according to your comments with following clarifications:

Comment1: “The GQDs are zero-dimension inorganic semiconductor...”. It should be ‘dimensional’.

Response: We have followed the reviewer’s recommendation. The word has been corrected as highlighted in the revised manuscript (Page 3).

Comment2: “One of the most types of humidity sensor is based on capacitive.”. The sentence seems incomplete.

Response: We have modified the sentence as highlighted in the revised manuscript (Page 4).

Comment3: “...longer lifetime, high responsibility, rapid response and recovery time and wider humidity detection range.”. Authors have discussed about ‘wider %RH range’ here, however in the examples they have provided detection ranges to be 35-60% and 20-90%. Please cite relevant examples with wide %RH range. E.g. 10.1039/C7TA02860A; 10.1039/C6NR08039A; 10.1039/C7TA05043G

Response: Thank you very much for suggesting the nice relevant papers. We have discussed and included the suggested references as highlighted in the revised manuscript (Pages 4, 5 and 19).

Comment4: The GQDs were synthesized by simple top-down via hydrothermal method and...” It is better to remove the word ‘top-down’ from here.

Response: We have followed the reviewer’s recommendation. This word has been removed as highlighted in the revised manuscript (Page 2 and 5).

Comment5: What is the standard deviation in Figure 5a

Response: The standard deviation for Figure 5 a is around $\pm 0.94\%$. The standard deviation has been included as highlighted in the revised manuscript (Page 9).

Comment6: Important %RH sensor based reference can be cited: doi.org/10.1063/1.5123479

Response: Thank you very much for suggesting the nice relevant paper. We have included the suggested reference as highlighted in the revised manuscript (Pages 4 and 18).

In addition, the English has been revised throughout the paper. The corrections have been made as highlighted using green color in the revised manuscript. We hope that the revised version based on recommendations is now suitable for publication. Thanks again for your valuable time helping to improve this paper.

Reviewer # 2:

Thank you for your review and valuable recommendation. The paper has been revised according to your comments with following clarifications:

Comment1: How did you vary humidity? explain in detail in the experimental section.

Response: We have followed the reviewer's recommendation. The details for varying humidity from 25% to 95% have been included as highlighted in the revised manuscript (Page 7).

Comment2: What is flow rate in SCCM of air entering in the sensing chamber and humid air. Does it vary for different humidity testing? Flow rate affects sensor's resistance and sensing performance. Ideally flow rate should constant for all measurement.

Response: The water vapor generated from a bubbling system was used as a source of humidity and placed in a temperature-controlled heat bath. The relative humidity (RH) range from 25% to 95% was varied by adjusting two mass flow controllers (MFC1 and MFC2). The mass flow ratios to obtain a desired %RH are listed in Table S1. The total flow rate (MFC1 + MFC2) was fixed at 200 SCCM. The additional details have been included as highlighted in the revised manuscript (Page 7). The flow rate ratios of MFC1 and MFC2 to obtain a targeted %RH (Table S1) has been included in the Supplementary information.

Table S1: Flow rate ratios of MFC1 and MFC2 to obtain a targeted %RH in our sensor chamber

RH (%)	MFC 1 (SCCM)	MFC 2 (SCCM)
30	165	35
35	150	50
40	140	60
45	130	70
50	115	85
55	110	90
60	90	110
65	75	125
70	65	135
75	60	140
80	45	155
85	30	170
90	20	180
95	5	195

Comment3: Please write resistance values of sensor in dry air and different %RH in manuscript.

Response: We have followed the reviewer's recommendation. The resistance values of humidity sensor in dry air and different %RH have been included as highlighted in the revised manuscript (Page 10).

Comment4: Ideally sensors response should be linear but in your case change in response is not significant after >50%RH. what could be reason.

Response: Thank you very much for your comment. As shown in Fig. 5a, a change in response seems to be not significant after >50 %RH because the resistance of humidity sensor at 25%RH is so high (~51.1 MΩ) and Y-axis is linear scale. To clarify this point, the enlarged graph for clear observation at high humidity is displayed in Fig. S1. The humidity sensor can significantly detect high humidity (>50%RH). The resistance of humidity sensor follows a good exponential response

relationship in the range of 25 - 95% RH. It may cause from contributions of the Schottky barrier modulation rather than ohmic contact upon water molecule adsorption. The additional details have been included as highlighted in the revised manuscript (Page 10). Fig. S1 has been included in the Supplementary information.

Figure S1: Resistance of GQD/AgNPs as a function of RH (30-95%).

Comment5: Letter C in the temperature unit should be in capital. please correct figure 1 and also correct in manuscript.

Response: We have followed the reviewer's recommendation. Letter C has been corrected to be a capital letter.

Comment6: Please provide references to mechanism explained in figure 6. Also work function of graphene shown in figure 6 is incorrect. Draw initial fermi levels of both GQDs and Ag and how band bending occurs and forms Schottky barrier. It would be great if you add it in figure 6.

Response: Thank you very much for pointing out our mistakes. We have followed the reviewer's recommendation. The Figure 6c and 6d have been corrected and redrawn. Additional details and relevant reference have been included as highlighted in the revised manuscript (Page 11,12 and 21).

Figure 6: Sensing mechanism illustrations of (a) the first and (b) the second physisorbed layer of the water molecules. (c) Energy band diagrams of GQDs and AgNPs before contact and (d) formation of Schottky barrier with/without H₂O adsorption.

In addition, the English has been revised throughout the paper. The corrections have been made as highlighted using green color in the revised manuscript. We hope that the revised version based on recommendations is now suitable for publication. Thanks again for your valuable time helping to improve this paper.